# Antidepressant-like and Beneficial Effects of a Neoponcirin-Beta-Cyclodextrin Inclusion Complex in Mice Exposed to Prolonged Stress

**DOI:** 10.3390/ijms25158289

**Published:** 2024-07-29

**Authors:** Luis José López Méndez, Lucía Martínez-Mota, Julia Cassani, Lilian Mayagoitia-Novales, Gloria Benítez-King, Luis Enrique Becerril-Villanueva, Ana María Dorantes-Barrón, Noé Jurado-Hernández, Rosa Estrada-Reyes

**Affiliations:** 1Departamento de Sistemas Biológicos, Universidad Autónoma Metropolitana, Unidad Xochimilco, Ciudad de México 04690, Mexico; ljlopez@correo.xoc.uam.mx (L.J.L.M.); cassani@correo.xoc.uam.mx (J.C.); 2Laboratorio de Farmacología Conductual, Dirección de Investigaciones en Neurociencias, Instituto Nacionalde Psiquiatría Ramón de la Fuente Muñiz, Ciudad de México 14370, Mexico; luciamota@ciencias.unam.mx; 3Departamento de Etología, Dirección de Investigaciones en Neurociencias, Instituto Nacional de Psiquiatría Ramón de la Fuente Muñiz, Ciudad de México 14370, Mexico; mayagn@inprf.gob.mx; 4Laboratorio de Neurofarmacología, Subdirección de Investigaciones, Instituto Nacional de Psiquiatría Ramón de la Fuente Muñiz, Ciudad de México 14370, Mexico; bekin@inprf.gob.mx; 5Laboratorio de Psicoinmunología, Dirección de Investigaciones en Neurociencias, Instituto Nacional de Psiquiatría Ramón de la Fuente Muñiz, Ciudad de México 14370, Mexico; lusenbeve@inprf.gob.mx; 6Laboratorio de Fitofarmacología, Dirección de Investigaciones en Neurociencias, Instituto Nacional de Psiquiatría Ramón de la Fuente Muñiz, Ciudad de México 14370, Mexico; ana1967@inprf.gob.mx.com (A.M.D.-B.); njurado2000@inprf.gob.mx (N.J.-H.)

**Keywords:** antidepressant, flavanone, inclusion complex, stress, anxiety, cyclodextrin complexes

## Abstract

Neoponcirin causes anxiolytic-like effects in mice when administered intraperitoneally but not orally. Neoponcirin is non-water-soluble and insoluble in solvents, and in medium acid, it isomerizes, reducing its bioavailability. To improve the pharmacological properties of neoponcirin, we formed a neoponcirin complex with beta-cyclodextrin (NEO/βCD), which was characterized by FT-IR, UV-Vis, and NMR, and their solubility profile. We evaluated the antidepressant-like effects of NEO/βCD acutely administered to mice orally in the behavioral paradigms, the tail suspension (TST) and the forced swimming (FST) tests. We also analyzed the benefits of repeated oral doses of NEO/βCD on depression- and anxiety-like behaviors induced in mice by chronic unpredictable mild stress (CUMS), using the FST, hole board, and open field tests. We determined the stressed mice’s expression of stress-related inflammatory cytokines (IL-1β, IL-6, and TNFα) and corticosterone. Results showed that a single or chronic oral administration of NEO/βCD caused a robust antidepressant-like effect without affecting the ambulatory activity. In mice under CUMS, NEO/βCD also produced anxiolytic-like effects and avoided increased corticosterone and IL-1β levels. The effects of the NEO/βCD complex were robust in both the acute and the stress chronic models, improving brain neurochemistry and recovering immune responses previously affected by prolonged stress.

## 1. Introduction

Anxiety and depression are the mood disorders most prevalent worldwide, and the World Health Organization (WHO) estimates that 20–25% of adults suffer from one of these disorders, which often have high comorbidity, and 41.6% of individuals can show both anxiety and depression diseases in the same period [1]. WHO had foreseen that by 2020, anxiety and depression disorders would become the primary contributors to global disability, without factoring in the onset of the emerging pandemic. Likewise, prolonged stress can be a trigger of anxiety and depression disorders [2]. Currently, there exists a wide variety of drugs for treating these mental illnesses. Among these are selective serotonin reuptake inhibitors (SSRIs) and tricyclic (TCAs), which have both antidepressant and antianxiety actions, or anxiolytics such as benzodiazepines and their derivatives that may effectively relieve symptoms of mild and severe depression [3]. Despite the efficacy of these treatments, sometimes these may be interrupted because of their undesirable or adverse effects, mainly when used for a prolonged time [4]. Thus, it is necessary to find new alternatives in the treatment of anxiety and depression.

In this regard, flavonoid compounds have been extensively studied as an alternative to conventional anxiety and depression treatment. Flavonoids’ anxiolytic, sedative, and nootropic effects are mediated by neurochemical systems such as the serotonergic and GABAergic neurotransmitter systems [5]. Also, there is evidence of positive flavonoid effects on rodent depressive-like behavior. It has been shown that these molecules exhibit antidepressive actions through different mechanisms, including modulation of receptors or increase in neurotrophic factors implicated in the neurogenesis processes [6].

Specifically, glycosides and aglycones of flavonoids have been evaluated on behavior with controversial results. For example, CNS effects of flavonoids are observed only when they are intraperitoneal or intraventricular administered or emulsified with surfactants such as Tween in water or soy oil and, in several cases, dissolved in the toxic solvent DMSO. Flavonoids aglycones and their glycosides are poorly soluble in organic solvents or water, affecting their effectiveness [7]. Previously, we described the anxiolytic-like actions of neoponcirin (syn.: didymin) intraperitoneally administrated in single doses [5]. Neoponcirin (NEO) is the 2*S*-5-hydroxy-4′methoxyflavanone-7-O-β-glucopyranosyl-(1→6)-β-rhamnoside. It is one of the main constituents of *Clinopodium mexicanum* [8].



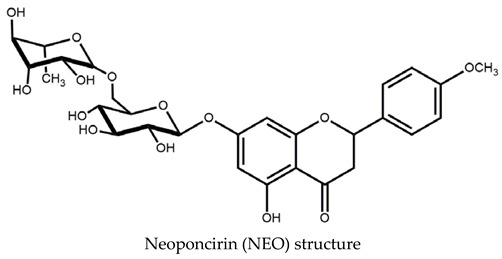



NEO’s low bioavailability limits its potential pharmacological effects, namely, its low solubility in water and organic solvents; coupled with its poor solubility, NEO precipitates in acid media, which favors its isomerization and decreases its effectiveness. Like many flavonoid glycosides, NEO is effective when intraperitoneally or intraventricularly administered. However, the oral treatment with NEO is ineffective in inducing anxiolytic-like effects [5], limiting its use in humans. This limitation can be overcome by forming inclusion complexes with macromolecules such as cyclodextrins. Cyclodextrins are macrocycles from oligosaccharides obtained from the starch glucosyltransferase degraded. These are valuable macromolecules that can be applied in the pharmaceutical field to improve the effectiveness of drugs for being products “generally recognized as safe” (GRAS) by the Food and Drugs Administration (FDA, USA) [9]. These carrier molecules have a three-dimensional shape resembling a truncated cone with different polarities in their interior and exterior surfaces [10,11,12]. They have a hydrophobic cavity that allows the host molecules to form inclusion complexes, improving their solubility and therapeutic effectiveness. This type of strategy has been shown to exist with other glycosides of flavonoids such as rutin, hesperidin, and hesperetin 7-*O*-glucoside [13,14]. In this study, we complexed NEO with β-cyclodextrin (NEO/βCD) to modify its physicochemical properties and evaluate its pharmacological profile in vivo after oral administration.

We analyzed the potential of NEO/βCD complex as an antidepressant treatment utilizing three different experiments. First, we evaluated the antidepressant-like effects of acute oral administration of NEO/βCD (4, 8, and 16 mg/kg) in the tail suspension (TST) and the forced swimming (FST) tests. In order to discard NEO/βCD nonspecific effects, the ambulatory activity of mice was measured in the open field test (OFT). To confirm the effectiveness of NEO/βCD complex in a model that resembles the depressive state, we analyzed the benefits of prolonged treatment with NEO/βCD once daily for 28 days on the behavior of mice kept under chronic mild stress. To assess the potential of NEO/βCD on immune and endocrine responses in experimental depression, changes in circulating stress-related proinflammatory cytokines (IL-1β, IL-6, and TNFα) and corticosterone levels of the stressed mice with CUMS were determined in serum. Results show that oral acute and repeated administration of NEO/βCD induced robust anxiolytic- and antidepressant-like effects without affecting the ambulatory activity of mice with or without under prolonged stress. NEO/βCD prevented some immunoendocrine alterations that are underlying the depressive-like behavior, evidencing complex interactions in the therapeutic response to NEO/βCD in stress-elicited states. NEO/βCD administered by oral route improved behavioral and psychoneuroimmunological markers associated with depression.

## 2. Results

### 2.1. Obtention and Analysis of the NEO/βCD Complex

#### 2.1.1. Analysis by Nuclear Magnetic Resonance (NMR) of NEO/βCD

Proton NMR (^1^H NMR) is a widely employed technique for comprehensively characterizing inclusion complexes (ICs). In scenarios where the host and the guest interact, distinct modifications in the chemical shifts of signals associated with interaction sites are observed [15].

Figure 1 shows a comparative analysis of the ^1^H NMR spectra of NEO/βCD IC and free NEO and βCD, highlighting the main signals that delineate the interaction between βCD and NEO within the inclusion complex.

Values corresponding to the main signal displacements are shown in Table 1.

The interactions governing complex formation are shown in Figure (inserts A, B, and C). Insert A reflects the chemical shifts of the signal attributed to the NEO, within the IC, specifically displacements of protons 6′ and 8′ in the ortho position between them within ring A at 6.35 ppm. Insert B shows the anomeric proton H-1′″ at 4.97 ppm and methyl protons of H-6′″ at 1.85 ppm from the rhamnose unit. Literature [13] suggests that the NEO structure likely could interact with βCD through rings A, C, and B of the aglycone, as well as the corresponding rutinoside moiety. The ^1^HNMR findings provide evidence of the inclusion complex formation, indicating that NEO flavonoid is enclosed within the βCD cavity, evidenced by chemical shifts displacements of protons H-3 at 3.63 ppm and H-5 at 3.55 from βCD (Insert C).

#### 2.1.2. Analysis by Ultraviolet-Visible (UV-Vis) and Fourier-Transform InfraRed (FT-IR) Spectroscopy Methods of NEO/βCD

In the UV-Vis spectrum (Figure 2A), a change in the maximum wavelength of NEO is observed from 282 nm to 289 nm. This bathochromic shift is explained by the change in the hydrophobic environment in the cyclodextrin cavity, which indicates NEO inclusion inside the βCD cavity. These results are agreed with those observed through NMR. Simultaneously, we observed a notable reduction in the intensity of the band at 340 nm, and the phenomenon may stem from the selective nature of the cyclodextrin interaction, further aligning with the streamlined signal patterns discerned within the NMR spectrum.

The NEO/βCD complex was also characterized using FT-IR, as shown in Figure 2B spectra of both free NEO and βCD were included as references. The observed changes in the distinctive bands related to the aromatic ring -C-H stretching vibrations of free NEO (at 1646 and 1511 cm^−1^) were tracked until their eventual disappearance. Additionally, the observed shift in the -C-O band at 1016 cm^−1^ within the FT-IR spectra provided conclusive evidence of the complex formation (Figure 2B).

#### 2.1.3. In Vitro Dissolution Study

The dissolution study showed a substantial difference between the dissolution profiles of NEO and the NEO/βCD complex. As shown in Figure 3, NEO alone exhibited a relatively low cumulative solubility of 3.46 × 10^−5^ ± 1.3 × 10^−5^ [mol/dm^3^], (12.93 ± 4.50%). In contrast, when combined with β-cyclodextrin (NEO/βCD), the solubility significantly improved, reaching a notably higher cumulative dissolution of 1.03 × 10^−3^ ± 4.84 × 10^−5^ [mol/dm^3^] (76.74 ± 3.619%). This considerable enhancement in dissolution indicates the effectiveness of the method used for the formation of the NEO/βCD complex, which improved the solubility and dissolution rate of NEO. However, we cannot discard the amorphization of inclusion complex contribution in this increase in solubility, presumably due to lyophilization [16]. 

### 2.2. Behavioral Evaluation of NEO/βCD

#### 2.2.1. Oral Acute Antidepressant-Like Effects of NEO/βCD in the TST and the FST

As shown in Figure 4A, the oral treatment with a single administration with increasing doses of NEO/βCD (4, 8, and16 mg/kg) produced a decrease in the duration of immobility concerning the control group and similar to clomipramine (CIM) at 25 mg/kg (H = 34.30, df = 4, *p* ≤ 0.001) in the TST. In this test, NEO/βCD reached their best effect at 16 mg/kg, which was significantly minor than the effect reached with CIM at 25 mg/kg. Similarly, acute oral doses of NEO/βCD (4, 8, and 16 mg/kg) and CIM (25 mg/kg; i.p.) caused a significant decrease in the immobility time in the FST. The effect was observed from 4 mg/kg dose to16 mg/kg (H = 39.11, df = 4, *p* ≤ 0.001), Figure 4B. In both tests, the most effective dose to reduce immobility time was 16 mg/kg of NEO/βCD, only the CIM effect was significantly greater than that caused by NEO/βCD.

It is important to mention that the antidepressant-like effects of drugs did not modify the counts number (H = 2.149, df = 4, *p* = 0.708) or the rearing number (H = 0.99, df = 4, *p* = 0.910) in the OFT, regarding the vehicle-treated group determinate in the OFT (Table 2). Thus, the reduction in immobility time induced by NEO/βCD was not associated with changes in ambulatory activity.

#### 2.2.2. Hole-Board and the FST to Measure the Effects of NEO/βCD on the Mice Subjected to Chronic Unpredictable Mild Stress (CUMS)

Figure 5 shows the effect of NEO/βCD and FLX treatment in the hole board test. 

As shown in panel A, NEO/βCD and FLX did not affect the mice’s ability to walk in such a way that no treatments caused significant changes in the ambulatory activity (count number; F_(3,24)_ = 0.43, *p* = 0.72), while NEO/βCD significantly increased the number of times that mice stood on their hind legs (rearing number; F_(3,26)_ = 5.32, *p* = 0.005) concerning to vehicle-treated group, panel B. NEO/βCD also significantly increased both the number and the time of head dipping (F_(3,24)_ = 7.13, *p* ≤ 0.001 and F_(3,25)_ = 32.55, *p* ≤ 0.001, respectively). At the same time, FLX did not cause changes in these behaviors (panels C and D). Notably, no significant difference was observed between NEO/βCD at 2 and 4 mg/kg treatments (t = −0.639, *p* = 0.5, t = 1.88, *p* = 0.10, respectively).

As shown in Figure 6, in mice maintained under stress for 28 days in the protocol of CUMS, higher levels of immobility time were observed in the FST. In addition, in chronically stressed mice, the concomitant oral administration once a day with NEO/βCD at 2 or 4 mg/kg/day or FLX at 2 mg/kg/day, i.p., significantly prevented the changes produced by chronic stress in the FST. Thus, mice exhibited reduced immobility time in the FST (F_(3,25)_ = 51.26, *p* ≤ 0.001) concerning the control group; FLX effectively prevented the effects of stress on immobility time, while non-differences were observed between FLX and NEO/βCD treatments.

All mice subjected to the FST under the acute or chronic treatment were tested in the open field test (OFT) to discard the possible changes of ambulatory spontaneous activity, which may evidence false negatives or false positives in these tests. As shown in Table 2, results showed that neither drug affected the ambulatory activity of mice. Thus, the decreased passive immobility behavior observed in the FST can be considered an antidepressant-like effect.

#### 2.2.3. Corticosterone and Interleukin Levels in the Serum of Mice Subjected to CUMS

At the end of CUMS protocol, the corticosterone from peripheral blood was measured by the ELISA method. As shown in Table 3, results showed that in the stressed group and treated with the vehicle, the corticosterone levels were superior to 1000 ng/mL, while in the groups treated with either NEO/βCD at 4 mg/kg (114.5 ± 4 ng/mL) or FLX at 2 mg/kg/day (139.5 ± 3 ng/mL), those were 10 times lower. Similarly, prolonged treatment with NEO/βCD and FLX significantly reduced the IL-1β levels of stressed mice in the group treated with the vehicle. Nevertheless, no differences between 2 and 4 mg/kg/day NEO/βCD were observed. The IL-6 levels were not modified with NEO/βCD at 4 mg/kg/day or FLX, while NEO/βCD at 2 mg/kg/day significantly increased it. Finally, none of the drugs significantly modified TNF-*α* levels.

For mice submitted to CUMS protocol and treated with the vehicle, the body weight remained constant in the basal measurement, possibly as a consequence of the prolonged stress. At the same time, FLX treatment significantly reduced the mice’s growth rate from day 5. This decrease remained until the end of the experiment. NEO/βCD treatment did not produce significant changes in the growth of mice; these data suggest that the complex reduced the effects of stressing stimuli in the growth of experimental subjects. However, no apparent toxicity signs were observed; the animals survived treatments and could resolve the behavioral challenges.

## 3. Discussion

Anxiety and depression are disorders that affect a large percentage of the world’s population. Depression and anxiety are multifactorial complex disorders, often comorbid. These public health problems can provoke social, working, and emotional disability, and severe cases can lead to death [1]. Drugs have been developed to relieve the symptoms associated with these disorders. However, more effective, faster-acting biomolecules with fewer side effects than current medications are still needed. 

In this context, flavonoids are shown to have interesting antidepressant and antianxiety properties, but their low bioavailability has limited their pharmacological potential. Although solid evidence shows the health benefits of these biodynamic molecules, the therapeutic outcome is still dependent on improving their profile effectiveness in oral administration. 

Different approaches have been developed to increase the absorption and bioavailability of poorly water-soluble substances such as flavonoids. An excellent method to resolve this controversy is coupling these with carrier agents, such as cyclodextrin, to form inclusion complexes [17]. Coupling biodynamic molecules with carrier agents to form inclusion complexes is an excellent approach. Thus, in this work, we used β cyclodextrins to form complexes that resulted in better absorption. These cyclic oligosaccharides have a hydrophobic core for hydrophobic guest molecules and a hydrophilic outer surface responsible for water solubilizing. Guest and host interact through intermolecular forces, which do not imply covalent binding but modify the physicochemical properties of the encapsulated guest [18]. 

In this work, we prepared a NEO inclusion complex (NEO/βCD) through the lyophilization method. The formation in a solution of the inclusion complex was analyzed through NMR and UV-Vis spectroscopies. As shown in Figure 1, the NMR analysis allowed us to make a first approach to the interaction modes between the host and the guest; we observed chemical shifts of the rhamnose unit of the NEO, protons H-1’”, and H-6’”, which are affecting it by interacting with βCD; this should be the moiety of the molecule that is into the cavity of the βCD [12]. These results indicate a first approach to how the NEO molecule interacts with the host. In the NMR analysis, we observed displacements in chemical shifts of the signals corresponding to aglycone moiety (Table 1), which is the most hydrophobic part of the molecule and is inserted into the cavity of the βCD.

UV-Vis analysis further demonstrated a bathochromic shift in the main band from 282 to 289 nm, suggesting enhanced stabilization of the excited state within the βCD cavity (Figure 2A). Subsequently, solid-state characterization of the complex was conducted using FT-IR, emphasizing only the detection of host bands in the complex spectrum as additional evidence of complexation (Figure 2B). 

In summary, the spectroscopic changes observed suggest the formation of a supramolecular NEO/βCD complex with physicochemical characteristics differing from free NEO. The formation of the inclusion complex can improve its solubility in water; it is well known that the amorphization caused by lyophilization of the inclusion complex results in high porosity in the product with larger surface areas for enhanced dissolution [16]. Thus, as shown in Figure 3, the solubility assay showed that NEO/βCD complexation facilitated the dissolution process and significantly increased the water solubility of NEO by 76.7% (1.03 × 10^−3^ ± 4.84 × 10^−5^ [mol/dm^3^]). This inclusion complex has a high degree of dispersion and increased solubility, which can facilitate the molecule to reach its pharmacological target in a biological system. 

As elaborated later, this modification positively influences its pharmacological effectiveness. According to this notion, the present study describes the in vivo pharmacological profile of NEO in the inclusion-complex form when administrated orally. We evaluated NEO/βCD effects in diminishing depressive-like behaviors in mice.

Evidence indicates that several types of stressors induce neurobiological changes and anxiety- and depression-like behaviors [19,20]. The experience of psychosocial stress and the inability to effectively deal with stress are widely recognized to contribute significantly to developing anxiety- and depression-like behaviors [21]. 

We tested the effects of oral acute and repeated administration with NEO/βCD in stress, depression, and anxiety paradigms in mice. 

In this first approach, our results showed that the treatment with single increasing doses of NEO/βCD (4 to 16 mg/kg) caused a significant reduction in the immobility time in both the TST and the FST, and these treatments did not modify the spontaneous ambulatory activity of mice in the OFT (Table 2). These results indicate that oral acute treatment with NEO/βCD induces a robust antidepressant-like effect without causing unspecific side effects. Also, it gives evidence that the NEO complexation with beta-cycle dextrin modified the physics and chemistry properties of the flavanone rutinoside encapsulated, improving its water solubility and absorption. The effects of the complex on the behavior allow us to suggest that NEO/βCD reached its final effector to exert the pharmacological actions on CNS. This study demonstrated certain specificity of the NEO/βCD on behaviors evoked by stress, which reflects signs of anxiety or depression. Such findings highlight the pharmacological viability of this complex and its implications in the therapeutic of affective disorders.

Sustained stress for a long time is one of the leading predisposing factors in developing anxiety and depression [22]. The capacity to cope with adverse situations is fundamental to survival and adaptation to the environment. When the environmental conditions unpredictably vary in humans, different strategies are implemented to cope with new environmental challenges. These adaptive responses involve physiological, behavioral, and neuroendocrine changes to recover homeostasis and the healthy functions of the organism. Furthermore, the sustained environmental stress triggers immune system dysfunction and hyperactivity of the hypothalamus–hypophysis–adrenal axis (HHA), with a consequence of hypercholesterolemia and mood disorders such as anxiety and depression [23].

Like humans, rodents have many behavioral resources to cope with abrupt environmental changes, and an active and passive response can be alternated to face these challenges. The CUMS is a paradigm that can cause of development of behaviors related to anxiety and depression in rodents [24]. This paradigm meets the face, hypothetical, and predictive validity for its use in the investigation of the neurobiological basis and therapeutics of affective disorders. Behavioral responses in animals submitted to CUMS are usually accompanied by neuroendocrine and immunological alterations [25]. Therefore, we explore the possible benefits of a treatment with NEO/βCD at repeated doses in mice under prolonged stress conditions. NEO/βCD oral treatment was administered once daily (for 28 days) to improve the adaptive responses of mice subjected to a paradigm known by its unpredictability. The concomitant administration of the treatment during the period of stress allows us to observe the potential of the complex to prevent the expression of depressive- or anxiety-like behaviors. Our results showed that animals subjected to CUMS and treated with repeated administration of NEO/βCD at 2 and 4 mg/kg/day had a significant reduction in immobility time in the FST when compared to animals without treatment. Still, similar immobility levels to those caused by the antidepressant selective serotonin reuptake inhibitor (SSRI), FLX (2 mg/kg/day), in the FST were observed. The effects of NEO/βCD on the levels of anxiety of mice subjected to CUMS were also evaluated at the end of the stress period in the HBT. As in the case of depressive behavior, NEO/βCD prevented the high levels of anxiety that characterize mice stressed with CUMS, since the complex increased all the exploratory behaviors evaluated, such as the number of times that mice stood on the hind legs (rearing number), and the exploration on holes, as well as the time spent. Such an increase in exploratory variables with the complex treatment was expressed in a more consistent form than the FLX, with no significant difference observed between 2 and 4 mg/kg/day of NEO/βCD, while neither treatment affected the spontaneous ambulatory activity, which supports the anxiolytic effects. These effects were comparable also to those produced with FLX (2 mg/kg/day) treatment. It is important to note that the effects elicited by a single intraperitoneal administration of NEO on affective-like behaviors [5] were also expressed when the NEO/βCD complex was administered in a schedule of prolonged oral treatment, improving the resilience of individuals in adverse situations. These results showed that such resilience was expressed in preventing the development of anxious-like and depressive-like behaviors and avoiding the hyperactivation of the HHA produced by prolonged stress. Interestingly, higher doses of the complex produced similar results on corticosterone levels than FLX, highlighting the translational value of the complex in the therapeutic of depression. It is possible that the CUMS exposure has led to the elevation of corticosterone levels by loss of the negative feedback from glucocorticoids, which leads to hypercortisolemia in humans and corticosterone in rodents. These findings suggest that the NEO/βCD in a similar form to antidepressants directly has produced increases in the expression or function of corticosteroid receptors in the brain, thus enhancing the negative feedback and reducing HHA axis activity, restoring axis homeostasis. Summarizing the evidence suggests that the beneficial effects of NEO/βCD observed in the mice’s behavior are partially produced by improving the regulation of the HHA axis and reducing the vulnerability of mice to stress.

Evidence indicates that both peripherical and central immune markers, such as proinflammatory interleukins, are altered after chronic stress and support the relationship between stress-induced increases as triggers of anxiety and depression diseases [21]. Even though inflammatory cytokines alterations are not specific to depression, there is also evidence that they are overexpressed in dysthymic disorders, such as generalized anxiety and depression, and IL-1β is the main mediator in the acute phase [26]; while prolonged stress causes activation of a series of receptors, such as monoaminergic, glutamate, and neuropeptide systems, and decreases in growth factors, such as brain-derived neurotrophic factor, which in turn also activate the transcription factors NF-kB and TNFα, increasing levels of IL-1β and IL-6, among others [27]. Our results showed that NEO/βCD treatment avoided the increase in IL-1β levels resulting from exposure to prolonged stress, notably with effects more robust at the higher dose; at the same time, the IL-6 levels remained high, while NEO/βCD did not significantly modify the TNF-α levels. Findings suggest that the complex prevented the activation of the immune system caused by chronic stress, which may contribute to the resilience of mice when they are evaluated in the anxiety and depressive tests. Prolonged treatment with NEO/βCD seems to stimulate adaptative responses to cope with adverse conditions. These allow us to suggest the NEO/βCD stimulates the immune and adaptative response [26].

The inflammatory cytokines IL-1β and IL-6 can also induce hyperactivity of the HHA axis [28] by altering the negative feedback of corticosteroids on said axis, inducing resistance of glucocorticoid receptors at the hypothalamus and pituitary levels [29], which in turn leads to behavioral changes, causing sick behaviors [29]. This mechanism of control of proinflammatory cytokines, particularly on the IL-1β, may explain at least in part the mechanism by which NEO/βCD (and FLX) regulates the corticosterone release, especially with the higher dose of the treatment. 

Overall, these results suggest that NEO/βCD treatment slows cellular deterioration, reduces inflammation, improves the indicators of the immune response, and restores the homeostasis of the HHA axis. The benefit of treatment is reflected in the decrease in the expression of depressive and anxious behaviors that contribute to preventing the establishment of deterioration caused by stress.

Finally, previously, we described the anxiolytic-like effects of intraperitoneal NEO and their association with the GABA complex system. In this regard, the GABA–benzodiazepines complex modulates anxiety levels; in particular, GABA-A receptors mediate fast synaptic inhibitory neurotransmission in the CNS [30]. However, although the GABAergic system participation is predominant in anxiety status, this also has a crucial role in some depression disorders [31]. Evidence indicates a role for GABA-A ionotropic receptors in physiologically modulating anxiety levels and depression-related behaviors [32,33]. Furthermore, corticosterone and stressors modify the pharmacodynamic properties of brain GABA-A receptors [34] and active biomolecules may act as positive allosteric modulators of these receptors, improving resilience [35]. Thus, it is possible to suggest that GABAergic mechanisms may participate in both anxiolytic- and antidepressant-like actions NEO/βCD. It is known that the interleukins in the brain also stimulate the secretion of norepinephrine, serotonin, and dopamine, which can positively modulate anxious and depressive states. Besides the GABAergic system, our findings do not discard the possibility that these neurotransmitter systems also participate in the NEO/βCD actions. The aforementioned allows us to continue the study of this exciting molecule.

## 4. Materials and Methods

### 4.1. Neoponcirin Isolation, Formation, and Characterization of the NEO/βCD Complex

Neoponcirin (NEO) was isolated from *Clinopodium mexicanum* ethanol extract, and their structural identity was determined by ^1^H, NMR spectroscopy, and comparison with literature data; for details, see Cassani et al., 2013 [5].

#### 4.1.1. Formation of the NEO/βCD Complex

An aqueous solution of βCD (15 mM) and an equimolar amount of NEO, previously diluted in methanol (1 mL), were mixed in a beaker. The mixture was kept under stirring for 4 h at 50 °C. Subsequently, the resulting suspension was filtered and lyophilized (−50 °C and 0.001 mBar) to obtain the NEO/βCD complex in the solid state. The complex was characterized by ultraviolet-visible (UV-Vis) spectroscopy, FT-IR, and nuclear magnetic resonance (^1^H NMR).

UV-Vis spectral analysis was performed using a VELAB-5100UV spectrophotometer with a quartz cell with a path length of 1 cm.

Fourier-transform infrared (FT-IR) spectra were recorded on a Thermo Scientific Nicolet 6700, in ATR mode, and the scanning range was 450−4000 cm^−1^ with 32 scans.

#### 4.1.2. NMR Study of the NEO/βCD Complex

^1^H NMR spectra were carried out at 600 MHz on an Agilent One Probe; both the NEO and NEO/βCD samples were analyzed at a concentration of 12 mM in DMSO-*d_6_* with temperature controlled at 298 K and acquired in non-spin mode. The difference between the chemical shifts of free NEO and NEO/βCD was calculated according to the following equation: Δδ = δ_complex_ − δ_free_.

#### 4.1.3. Dissolution Study

A dissolution assay was conducted to assess the release profile of NEO and its β-cyclodextrin complex (NEO/βCD) in distilled water. In conical tubes, 1.2 mg of pure NEO and an equivalent amount of NEO/βCD in a 1:1 ratio were dissolved in 1.5 mL of distilled water. The tubes were placed in a thermomixer, maintaining a temperature of 25 °C with constant agitation at 100 rpm throughout the study. Samples were collected at specified time intervals (10, 15, 30, 45, 60, 90, 150, and 240 min), followed by centrifugation at 3000 rpm for 3 min. A 600 µL aliquot from each centrifuged sample was analyzed via UV-Vis spectrophotometry. Each sample was carried out in triplicate. Concentration determination was achieved by interpolating the absorbance at 284 nm against a previously constructed calibration curve for NEO and NEO/βCD. The cumulative solubility was calculated with the following equation [36]:$${Cumulative\;solubility}_{t}\left(mol/{dm}^{3}\right)=P_{t-1}+P_{t}$$
where

$P_{t}$ Neo concentration at a time “*t*”

$P_{t-1}$ Neo concentration previous to “*t*”

### 4.2. Pharmacological Evaluation

#### 4.2.1. Animals

Male Swiss Webster mice (25–30 g) were maintained under inverted 12 h light/dark cycle conditions at 22 ± 1 °C temperature, housed in eight per group with free access to water and food (Purina Chow, by LabDiet of PMI Nutrition International, LLC, 4001 Lexington Avenue North Arden Hills, MN 55126, EE. UU. Animal experiments followed NIH guidelines for the care and use of laboratory animals, 8th edition (2011) [37] and the Mexican legislation (NOM-062-ZOO-1999) of the Secretary of Agriculture, Livestock, Rural Development, Fisheries, and Food (SAGARPA, Camino a Nativitas S/N, Barrio Xaltocan, Ciudad de México. C.P. 16090, México). The local ethical committee approved this study and all animal experimentation procedures (NC19054; 2019–2024). Mice were tested between 9:00 and 14:00 in a room lit by dim light. All experiments were video recorded for later analysis.

#### 4.2.2. Drugs

NEO/βCD solutions were prepared daily (10 mL/kg). Control groups were treated with the vehicle containing a β-cyclodextrin (βCD, Sigma, Life Science, Sigma-Aldrich, Co., 3050 Spruce Steet, St. Louis, MO, USA) (15 mM) in water solution. Fluoxetine (FLX) and clomipramine (CIM) were purchased from Sigma Life Science (Sigma-Aldrich, Co., 3050 Spruce Steet, St. Louis, MO, USA).

#### 4.2.3. Tail Suspension Test (TST)

This device consisted of a wooden rod (2 × 2 cm) placed 50 cm above a sawdust bed, to which the mouse was fastened by the tail 1 cm from the tip with adhesive tape. Each mouse was suspended for 6 min, and long-lasting immobility was measured for the final 4 min of the assay. Immobility time was scored when the mouse remained passive and completely motionless. The accumulated immobility time (sec) reduction is considered an antidepressant-like effect [38].

#### 4.2.4. Forced Swimming Test (FST)

Mice were individually placed into glass cylinders (height: 21 cm, diameter: 14.5 cm) containing 15 cm of water at 23 ± 2 °C. Two swim sessions were conducted: the first 15 min (pre-test) to promote the immobility behavior and 24 h later, the second 5 min (test). At the end of each session, the animals were removed, gently dried, placed in a cage to recover (23 ± 2 °C) for 15 min, and returned to their home cages. The test session was videotaped and later scored by an observer blinded to the treatments applied.

#### 4.2.5. Open Field Test (OFT)

OFT consisted of an opaque Plexiglas box (40 × 30 × 20 cm) with the floor divided into 12 equal squares. At the start of the test, the animal was placed in the corner of the box. Their behavior was videotaped for 5 min to register the number of times the mouse crossed from one square to another (count number) and the number of times it stood on its hind legs (rearing number) [39].

#### 4.2.6. Hole Board Test (HBT)

The hole board device consists of a 60 × 30 × 15 cm wooden box with four equidistant holes (1 cm diameter) on the floor divided into four equal quadrants. At the start of the test, the mouse was placed at the center of the board, illuminated by indirect and dim light. Each animal was put in the center of the hole board and allowed to be freely explored for 5 min. The number, time spent head-dipping, and the rearing number (when a mouse stands up on their hind legs) were recorded. The board floor was carefully cleaned and dried between trials with alcohol (ammoniacal (1%) in 70% ethanol solution) wipe to remove traces from the previous test. An increase in head dipping number, head dipping time, and the rearing number compared to the control group is considered an anxiolytic effect. In contrast, decreasing these behaviors is considered a sedative effect [5,40].

#### 4.2.7. Evaluation of the Oral Acute Effects of NEO/βCD in Despair Tests

Experimental Design

Mice housed in independent groups of eight each were treated with a single oral dose of saline (CTL), or the NEO/βCD at 4, 8, or 16 mg/kg, or intraperitoneally dose of clomipramine at 25 mg/kg, and 30 min after were submitted to TST, FST, or OFT [41].



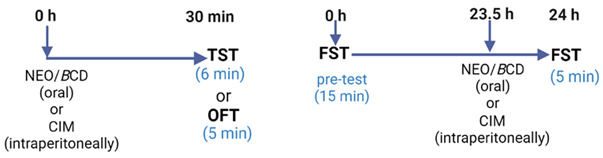



#### 4.2.8. Evaluation of the Chronic Effects of NEO/βCD in Animals Subjected to the Chronic Unpredictable Mild Stress Test (CUMS)

The CUMS protocol consisted of various unpredictable mild stressors applied once a day for 25 consecutive days. Mice (housed in groups of eight) received different adverse stimuli that were chosen randomly, without a set schedule, and they could not predict them. These stimuli were not employed more than two times at the same day/night cycle time. The mice were exposed once a day to the following adverse stimuli: 6 h of body restraint, 2 h of food deprivation, 3 h of water deprivation, 12 h of overnight illumination, 12 h of wet padding, and 24 h of the day/night inversion (Willner, 2016; Arrieta Báez et al., 2022 [26]). To corroborate the effects of the stressors, on day 28, animals were subjected to the OFT and 5 min after to the HBT, and the next day (29 days), they were subjected to a swim session of 15 min (pre-test), and 24 h later, they were subjected to the FST (test; 5 min) and 5 min after, to the OFT. Independent groups of 8 animals each were treated with saline, NEO/βCD at 2, or 4 mg/kg/day, or fluoxetine (FLX; 2 mg/kg/day); another group was kept the same conditions of feeding and housing but received no stressors stimuli, served as control. All groups were administered 30 min before receiving the stressful stimuli. 



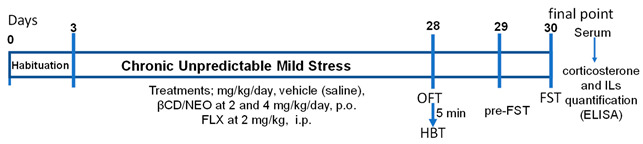



To evaluate the antidepressant-like effect of NEO/βCD in animals subject to chronic stress on the last day of the CUMS test (28 days) and 30 min after the last administration, the mice were submitted to the swim session of 6 min. The session was video-recorded, and the passive behavior was measured as the immobility time (sec) during the final 4 min.

Timeline of the chronic experimental design using the CUMS procedure and a chronic administration of NEO/βCD. Habituation was carried out in the first three days. CUMS and treatments were applied for 25 days.

Evaluations were conducted during the last three days of treatments: chronic unpredictable mild stress (CUMS), hole board test (HBT), forced swimming test (FST), fluoxetine (FLX), and neoponcirin-β-cyclodextrin complex (NEO/βCD).

The CUMS stress produces depression-like behaviors expressed as increases in the passive behavior of immobility in the FST.

At the end of the experimentation, TNF*α*, IL-1β, IL-6, and corticosterone levels in peripherical serum were measured by the ELISA method [24].

Mice were quickly sacrificed by decapitation 5 min after the FST; blood was collected in a vacutainer tube previously sprinkled with heparin at (10 UI/mL) and centrifugated at 8000 G for 15 min at −20 °C; the serum was separated into aliquots and stored at −40 °C until subsequent analysis. 

Corticosterone was quantified in aliquots of 10 µL of serum using an enzyme-linked immunosorbent assay (ELISA) kit (ENZO;10 Executive Blud., Farmingdale, NY, USA); IL1β and IL6 were quantified using ELISA kit (INVITROGEN; Bender Med Systems GmbH, Campus Vienna Biocenter 2 A-1030, Vienna, Austria and ENZO;10 Executive Blud., Farmingdale, NY, USA, respectively). Quantifications were performed following the manufacturer’s instructions.

### 4.3. Statistical Analysis

Statistical analysis and figures were performed using Sigma Plot software (13.2 version). Differences between groups were analyzed using parametric or non-parametric one-way analysis of variance (ANOVA), and mean values were considered statistically significant at *p* ≤ 0.05, following the pairwise multiple comparison procedures (Holm-Sidak or Mann–Whitney Ran Sum tests). All data are shown as mean ± SEM. In figures, the significance levels are denoted as * *p* < 0.05, ** *p* < 0.01, *** *p* < 0.001.

The study followed the Declaration of Helsinki and was approved by the Ethics Committee of Instituto Nacional de Psiquiatría Ramón de la Fuente Muñiz, protocol number CEI/NC19127.0.

## 5. Conclusions

Acute oral administration with NEO/βCD caused antidepressant- and anxiolytic-like effects in mice. The repeated oral administration of NEO/βCD produced positive benefits in mice under prolonged stress, causing anxiolytic and antidepressant-like effects, avoiding the corticosterone elevation and the IL1β proinflammatory overexpression.

The results suggest that the formation of the NEO/βCD complex holds promise for enhancing the therapeutic performance of NEO by improving its solubility, thereby potentially overcoming challenges associated with its limited aqueous solubility and facilitating its administration.

## Figures and Tables

**Figure 1 ijms-25-08289-f001:**
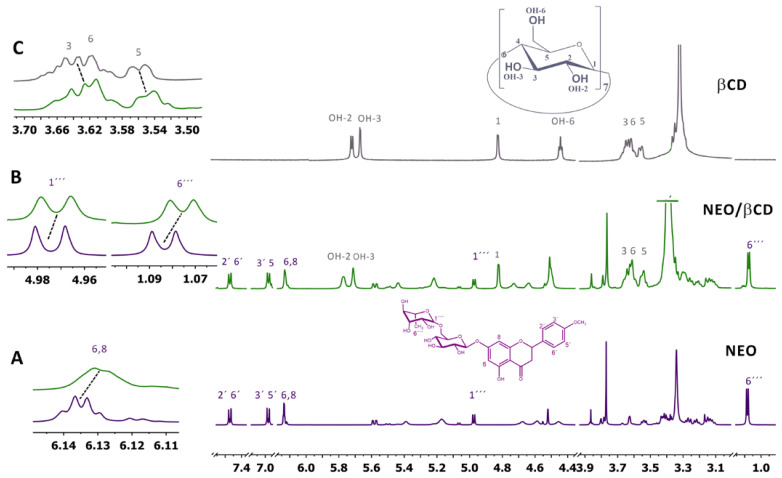
Comparison of ^1^H NMR spectra profile displacement of NEO/βCD (neoponcirin/beta cyclodextrin Inclusion Complex; IC) concerning NEO (neoponcirin). Drugs were dissolved in DMSO-d6. (**A**) ---NEO free, (**B**) ---IC complex, and (**C**) ---βCD free.

**Figure 2 ijms-25-08289-f002:**
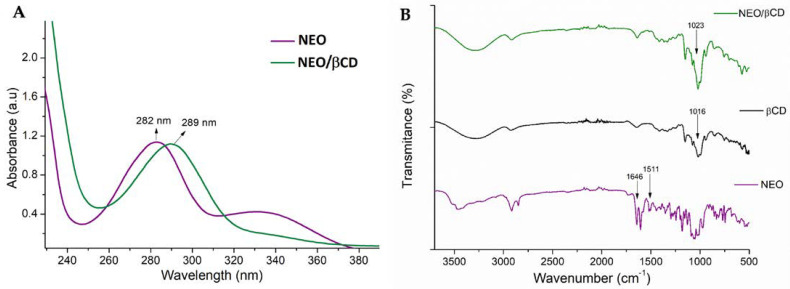
UV-Vis absorption spectra of NEO/βCD in comparison with NEO (**A**), FT-IR spectra of NEO (purple), βCD (black), and βCD/NEO complex (green) (**B**).

**Figure 3 ijms-25-08289-f003:**
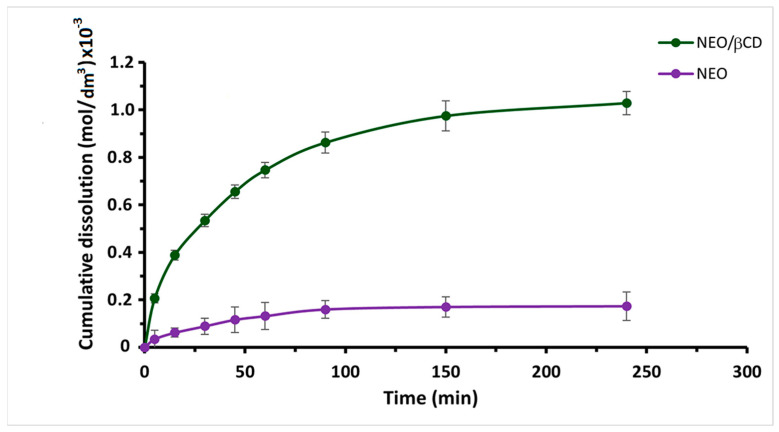
Drug dissolution profiles of NEO and the NEO/βCD complex. The points represent the average over three repeats. Remarkably, the cumulative dissolution of NEO reached 12.93% ± 4.506%, whereas the dissolution for NEO/βCD notably exceeded this at 79.74% ± 3.619%. These findings unequivocally highlight a significant increase in the complex’s solubility and dissolution rate concerning the NEO.

**Figure 4 ijms-25-08289-f004:**
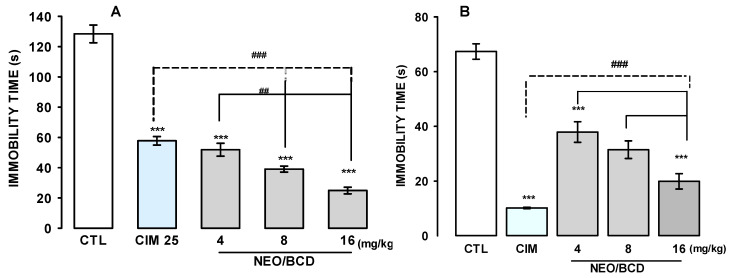
Effects of a single oral dose of NEO/βCD in the tail suspension (**A**) and the forced swimming (**B**) tests. A, NEO/βCD, CTL, control group; CMI 25, clomipramine at 25 mg/kg; and NEO/βCD, complex at different tail suspension test (TST) doses.B control group; CMI 25, clomipramine at 25 mg/kg; NEO/βCD, complex at different doses in the forced swimming test (FST). Data represent the mean ± standard error of the mean of the accumulated immobility time of groups of 8 to 10 mice each. Differences between treatments were analyzed with non-parametric Kruskal–Wallis’ one-way ANOVA (*** *p* ≤ 0.001) and pairwise multiple comparison procedures (Dunn’s method). Dunn’s method comparison: NEO/BCD 4 (mg/kg) vs. NEO/BCD 16 (mg/kg); ^##^ *p* ≤ 0.01, and: CIM vs. NEO/BCD 16 mg/kg; ^###^ *p* ≤ 0.001.

**Figure 5 ijms-25-08289-f005:**
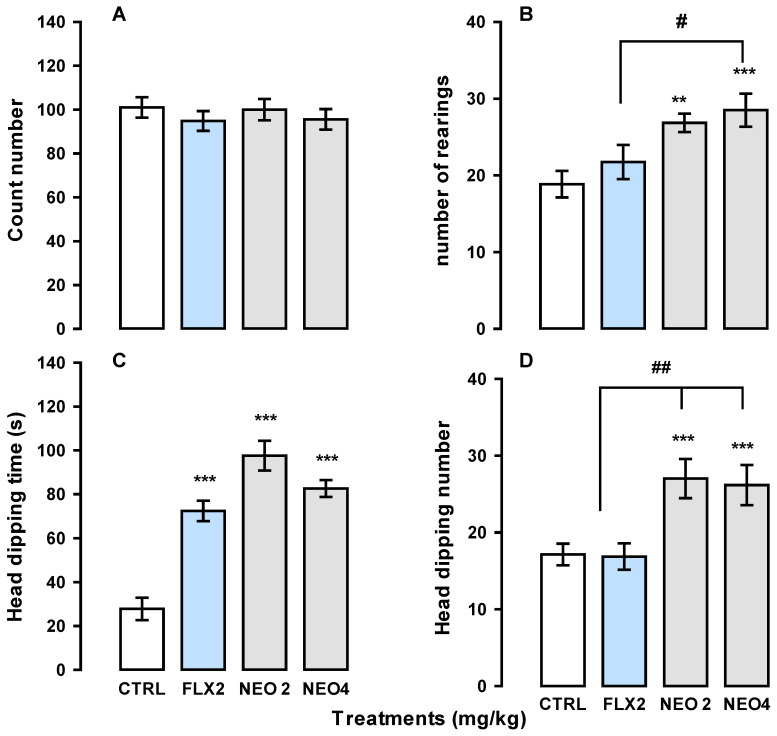
Effect of the complex NEO/βCD on mice exposed to chronic unpredictable mild stress and evaluated in the hole board test. Graphics represent (**A**) the count number, (**B**) rearings number, (**C**) head dipping time (s) and (**D**) the head dipping number. CTL, control group; FLX 2, fluoxetine at 2 mg/kg/day; NEO2, NEO/βCD at 2 mg/kg/day; NEO4, NEO/βCD at 4 mg/kg/day. Data represent the mean standard error the number of events (**A**,**B**,**D**) or seconds (**C**), of groups of 7 or 8 mice, each one. Treatment effects was analyzed with one-way analysis of variance (ANOVA), following Holm-Sidak’s method for pairwise multiple comparisons. Statistical significance is indicated as follows: ** *p* ≤ 0.01, *** *p* ≤ 0.001 versus the vehicle-control group; ^#^
*p* ≤ 0.05, ^##^
*p* ≤ 0.01 FLX vs. NEO2 and NEO4.

**Figure 6 ijms-25-08289-f006:**
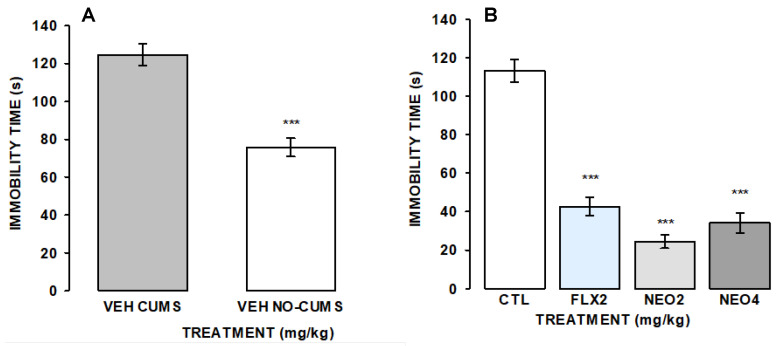
Effect of NEO/βCD on the FST of animals exposed to chronic unpredictable mild stress (CUMS). (**A**) Animals administered with vehicle (VEH CUMS) and exposed to CUMS (grey bar) and animals administered with the vehicle without CUMS (VEH NO-CUMS; white bar). (**B**) All animals’ groups were exposed to CUMS and tested in the FST at the end of the treatments. CTL, control group; FLX 2, fluoxetine at 2 mg/kg/day; NEO2, NEO/βCD at 2 mg/kg/day; NEO4, NEO/βCD at 4 mg/kg/day. Data represent the mean standard error of immobility time of groups of 7 or 8 mice, each one. Treatment effect was on depressive-like behavior was analyzed with one-way analysis of variance (ANOVA) followed by the Holm-Sidak’s method for pairwise multiple comparisons, *** *p* ≤ 0.001 versus the vehicle-control group.

**Table 1 ijms-25-08289-t001:** ^1^H NMR chemical shifts of the main protons with significant changes (Δδ).

^1^H NMR	Chemical Shifts (ppm)			^1^H NMR	Chemical Shifts (ppm)		
βCD		NEO/βCD	Δδ = δIC − δβCD	NEO		NEO/βCD	Δδ = δIC − δNEO
H1	4.824	4.823	−0.001	H6, H8	6.137	6.127	−0.009
H3	3.601	4.593	−0.007	H1′″	4.976	4.790	−0.186
H5	3.662	4.643	−0.018	H6′″	1.082	1.068	−0.014

**Table 2 ijms-25-08289-t002:** Effect of NEO/βCD, CIM, and FLX on the ambulatory activity.

Acute treatment (mg/kg)	Count number	Rearing number
CTL	48.37 ± 2.50	30.25 ± 1.69
CIM 25	55.25 ± 2.25	27.75 ± 2.11
NEO/βCD 4	51.87 ± 4.90	30.25 ± 1.65
NEO/βCD 8	45.50 ± 4.27	28.25 ± 1.53
NEO/βCD16	46.00 ± 4.03	30.25 ± 2.05
	F_(4,35)_ = 1.20, *p* = 0.32	F_(4,49)_ = 0.46, *p* = 0.76
Chronic treatment (mg/kg/day)	Count number	Rearing number
CTL	52.12 ± 2.91	28.87 ± 1.77
FLX 2	47.00 ± 4.90	24.12 ± 2.25
NEO/βCD 4	49.62 ± 3.07	30.12 ± 2.01
NEO/βCD 8	50.60 ± 4.46	29.62 ± 1.78
	F_(3,28)_ = 0.30, *p* = 0.82	F_(3,28)_ = 1.96, *p* = 0.14

Data represent the mean ± standard error of the mean of independent groups of eight animals. Differences between groups were measured by one-way analysis of variance (ANOVA).

**Table 3 ijms-25-08289-t003:** Effect of oral chronic treatment of NEO/BCD and FLX on corticosterone, IL-1β, and IL-6 of mice subjected to CUMS.

Treatment (mg/kg/day)	Corticosterone (ng/mL)	IL-1β (pM/mL)	IL-6 (pM/mL)
VEH	1159.14 ± 184.83	798.83 ± 63.49	258.57 ± 20.01
FLX 2	139.50 ± 33.06 **	567.83 ± 42.16 **	291.55 ± 20.48 ^#^
NEO/βCD 2	303.14 ± 50.54 **	624.83 ± 28.63 *	387.43 ± 35.80 ^#^
NEO/βCD 4	125.50 ± 32.55 **	587.29 ± 25.27 **	280.57 ± 17.37
	H = 17.48, df = 3, *p* = 0.002	F_(3,20)_ = 6.11, *p* = 0.004	F_(3,20)_ = 6.27, *p* = 0.005

Data were analyzed using a one-way ANOVA, with an overall significance level =0.05, followed by all pairwise multiple comparison procedures of the Holm-Sidak method. Decrease: * *p* ≤ 0.05, ***p* ≤ 0.01; increase: ^#^
*p* ≤ 0.05.

## Data Availability

The data presented in this study are available on request from the corresponding author.

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
