# Peer review of "Antidepressant-like and Beneficial Effects of a Neoponcirin-Beta-Cyclodextrin Inclusion Complex in Mice Exposed to Prolonged Stress"

_ijms, 2024, doi:10.3390/ijms25158289_

Round 1

Reviewer 1 Report

Comments and Suggestions for Authors

General comment:

This manuscript, entitled “Antidepressant-like and beneficial effects of a neoponcirin βcyclodextrin inclusion complex in mice exposed to prolonged stress,” authored by Méndez et al., investigated the role of cytochrome p450 reductase in metabolic activity by CYP6P9a/-b. In this manuscript, the author puts effort into studying the solubility profile of neoponcirin encapsulated in beta cyclodextrin by biophysical tools. The work has good potential to apply this method for hydrophobic drug delivery to increase its efficiency. After answering the following questions, this work can be considered for publication in the International Journal of Molecular Sciences.

Specific comments:

1)      The use of beta-cyclodextrin to increase the solubility of hydrophobic compounds is well established; why does the author think it is valuable for other unknown future applications?

2)      How does the author distinguish encapsulated neoponcirin into beta-cyclodextrin or hanging superficially or bonded but hanging on a beta-cyclodextrin donut surface?

3)      The UV vis signal for free neo in the range of 320 to 380 decreased in neo/Bcd. Does the drug interact with beta-cyclodextrin? Although the NMR signal showed no chemical interaction, that may be because of the superficially hanging neo on the surface.

Comments on the Quality of English Language

minor editing

Author Response

We are very grateful for the peer review of the manuscript. Your comments have greatly enriched our work and knowledge on the topic, and we greatly appreciate the time you have dedicated to this paper.

We have addressed all your observations in the manuscript, which we are explaining in the following point-to-point letter. To facilitate the review of the text, the changes in the manuscript are marked in red.

Review 1

1)The use of beta-cyclodextrin to increase the solubility of hydrophobic compounds is well established; why does the author think it is valuable for other unknown future applications?

Answer

Different natural products or their derivatives have problems to reach their therapeutic targets in in vivo models. Through enhancing the solubility of neoponcirin, we aim to unlock its therapeutic efficacy more effectively. Our research endeavors to expand upon this by delving into its potential in treating mood disorders.

Usually, natural products, especially flavonoid supplements available on the market do not have bioavailability studies before they are released to the market, so their effectiveness is questionable or unknown, giving divergent results or associated with placebo effects. Our study put together the use of inclusion complexes, the evidence shown on solubility of the complexed combination, and the results of its pharmacological actions in a long-term study, supports the use of this kind of complexes as an adequate (potential) pharmaceutical formulation both at an experimental level and for possible translation to future clinical use. Furthermore, the intraperitoneal route produces serious irritation problems associated with generalized inflammation and, in some cases, causes abdominal pain. In contrast, the oral route is less irritating at the level of the peritoneum and does not cause inflammation, especially in long-term treatments. Unlike the intraperitoneal route usually used in experiments with mice, the oral route is the most simple and common administration route in humans.

Additionally, flavonoids are biologically active, and they generate great interest for their potential applications in functional foods and pharmaceuticals. Our study can provide valuable insights for developing new pharmaceutical forms.

2)      How does the author distinguish encapsulated neoponcirin into beta-cyclodextrin or hanging superficially or bonded but hanging on a beta-cyclodextrin donut surface?

Answer

Neoponcirin may be interacting with both inside or outside the cavity of the βCD. It is known to be associated within the cavity due to the chemical shifts observed for the signals of protons H-3 and H-5. These chemical shifts were measured as we explained in the text, lines 128 to 131; and showed in the figure 1 (inset C).

3)      The UV vis signal for free neo in the range of 320 to 380 decreased in neo/Bcd. Does the drug interact with beta-cyclodextrin?

Although the NMR signal showed no chemical interaction, that may be because of the superficially hanging neo on the surface.

Answer

NMR is one of the most employed techniques to demonstrate how the host and guest interact in the formation of inclusion complexes. A simple review of the different inclusion complexes allows us to observe which technique best will enable us to describe whether the compound is located outside or inside the βCD cavity, by measuring the shifts of protons in the spectra between associatedβCD and βCD free (Δδ= δIC-δ βCD). In this paper, we report the differences in the chemical shifts between the CD free and CD in the complex, as well as NEO in the complex and NEO free (Δδ= δIC-δNEO). Table 1.

Sincerely,

Rosa Estrada-Reyes

Laboratorio de Fitofarmacología

Instituto Nacional de Psiquiatría Ramón de la Fuente Muñiz

Ciudad de México, Mexico

Reviewer 2 Report

Comments and Suggestions for Authors

The submitted manuscript describes the results of the experimental work in which the authors aimed to study the complexes formed between the beta cyclodextrin and neoponcirin (didymin), focusing on their anticipated biological activity. While I really appreciate the presentation style, figures and tables are nicely prepared, I have also some serious concerns about this work.

The Authors have studied previously [5] the effect of the neoponcirin on mice, and this work is the continuation of their studies, in particular how the addition of cyclodextrin increases the dissolution and, consequently, bioavailability of this API. Therefore, I don’t understand why the authors haven’t used a positive control group in which the NEO would be administrated at the same dosage, but non complexed, without CD. It is reasonable to have a negative control group with CDs and the authors have done this. But what about the positive control?

The other aspect I’m very worried about is the dissolution analysis. The Authors have lyophilized the obtained complexes, which led to the amorphization. It is well known that the amorphous (lyophilized) APIs are characterized by more rapid dissolution. On the other hand, NEO used for comparison was not lyophilized. Therefore, the Authors don’t know whether the increase in the dissolution rate was caused by the complexation or lyophilization or was the result of those both factors. To be honest, the Authors should perform two more dissolution studies: one with crystallized complexes (not subjected to lyophilization) and one with lyophilized NEO.

Line 41, instead of „beta dextrin-complexes” it should be “cyclodextrin complexes”

Line 49, but 2020 has already been 4 years ago… so past tense should be used

Line 74, what is “a rhamnoside de flavanone”?

In the introduction, the authors should include a figure presenting the structure of NEO

In the introduction, it should be clearly stated if there were any previous articles describing the complexes of NEO/βCD, i.e. [13], and what is currently known about such complexes

Line 118, here, after such statement, it would be beneficial to include a reference to a review on NMR analysis of cyclodextrin complexes

Table 1, chemical shifts should be rounded at most to 3 decimal places, due to the accuracy of this method

Figure 2A, spectrum of Beta-CD should be added to enable the proper comparison

Figure 3, the dissolution should be presented in mol/dm3 to show if really higher concentration of NEO can be achieved by the used of beta-CD.

Line 469, it should be “NMR” not “NM”

Lines 476-477, the details of FT-IR and UV-VIS analysis must be provided. I.e., the spectrometers used, resolution, mode (KBr? ATR?), similarly to how the

Lines 516-518, this sentence is incomplete

Lines 656-661, this part should be removed

Table 2, the values are not rounded properly, i.e. it should be 51.87 ± 4.90

Lines 308-309, this is not entirely true, the complexes don’t posses better absorption as CD is not being absorbed but metabolized. The formation of complexes

Lines 324-326, this statement about enantioselectivity is based on exactly which results?

Author Response

We are very grateful for the peer review of the manuscript. Your comments have greatly enriched our work and knowledge on the topic, and we greatly appreciate the time you have dedicated to this paper.

We have addressed all your observations in the manuscript, which we are explaining in the following point-to-point letter. To facilitate the review of the text, the changes in the manuscript are marked in red.

Reviewer 2

The submitted manuscript describes the results of the experimental work in which the authors aimed to study the complexes formed between the beta cyclodextrin and neoponcirin (didymin), focusing on their anticipated biological activity. While I really appreciate the presentation style, figures and tables are nicely prepared, I have also some serious concerns about this work.

1) The Authors have studied previously [5] the effect of the neoponcirin on mice, and this work is the continuation of their studies, in particular how the addition of cyclodextrin increases the dissolution and, consequently, bioavailability of this API. Therefore, I don’t understand why the authors haven’t used a positive control group in which the NEO would be administrated at the same dosage, but non complexed, without CD. It is reasonable to have a negative control group with CDs and the authors have done this. But what about the positive control?

Answer

Thank you very much for this observation. We can report that our previous study evidenced a robust anxiolytic effect of NEO when this was acutely administered (one dose) by intraperitoneally route. In turn, acute administration of one oral dose was unable to replicate the anxiolytic-like effect of NEO intraperitoneally administered. In that study, evaluation of free NEO on depressive-like behaviors (FST or TST) was not conducted. Stress triggers anxiety and depressive-like behaviors in mice, and usually, treatments and dosage reducing anxiety also reduce depressive-like behaviors (Martínez-Mota L, et al., 2020. Calea zacatechichi Schltdl. (Compositae) produces anxiolytic- and antidepressant-like effects and increases the hippocampal activity during REM sleep in rodents.

  1. Ethnopharmacology. 265, 30. https://doi.org/10.1016/j.jep.2020.113316 ISSN 1872-7573;  Cárdenas J, et al., 2017. Anxiolytic- and antidepressant-like effects of an aqueous extract of Tanacetum parthenium L. Schultz-Bip (Asteraceae) in mice. J. Ethnopharmacology. https://doi.org/ 10.1016/j.jep.2017.02.023).

We agree with the reviewer that the study has this limitation. However, based in our experience and interpretation of the study of anxiety (Cassini et al., 2013. Anxiolytic-like and antinociceptive effects of 2S -Neoponcirin in mice. Molecules, 18, 7584-7599. DOI: 10.3390/molecules18077584), animals treated with NEO (free) would not respond with reduced despair in the FST or TST, as they were not able to respond with reduced anxiety-like behavior in the hole board test.

While this evaluation remains to be conducted, on the previous interpretation, we conducted a pilot assay to determine the antidepressant-like doses of NEO/BCD, which were used in the present study. In all cases (acute or repeated treatments) negative control with BCD was included as a comparative group. 

2) The other aspect I’m very worried about is the dissolution analysis. The Authors have lyophilized the obtained complexes, which led to the amorphization. It is well known that the amorphous (lyophilized) APIs are characterized by more rapid dissolution. On the other hand, NEO used for comparison was not lyophilized.

Therefore, the Authors don’t know whether the increase in the dissolution rate was caused by the complexation or lyophilization or was the result of those both factors. To be honest, the Authors should perform two more dissolution studies: one with crystallized complexes (not subjected to lyophilization) and one with lyophilized NEO.

Answer

We understand your concern regarding the contribution of lyophilization to the increase in solubility. The evidence obtained through NMR, UV-Vis, and FT-IR presented in this article supports the formation of a complex between neoponcirin and cyclodextrin, which positively impacts their solubility.

We recognize that amorphization due to lyophilization may also contribute to the increase in solubility. This acknowledgment has been included in the analysis of results, specifically in the 2.1.3 section. However, the primary objective of this study was not to quantify the individual impact of amorphization and complexation, but rather to demonstrate an overall positive effect on the solubility of neoponcirin.

We isolated the neoponcirin as an amorphous solid by precipitation after the chromatographic procedure, for which we did not consider lyophilization necessary.

However, the chemical changes in UV and NMR joint to water solubility are a robust data of the formation of NEO complex. We agree with you, we will assay the crystallization the crystallization in future experiments.

We are very grateful for this comment and suggestions. We believe that additional studies can be addressed on differentiating the effects of amorphization and complexation, which will further enrich our understanding of the phenomenon.

3) Line 41, instead of „beta dextrin-complexes” it should be “cyclodextrin complexes”

Answer

This was corrected.

4) Line 49, but 2020 has already been 4 years ago… so past tense should be used

Answer

This was fixed.

5) Line 74, what is “a rhamnoside de flavanone”?

Answer

Thank you for your observation. This name was corrected, the complete name was included in the text to avoid confusion, in the present version say: Neoponcirin (NEO) is the 2S-5-hydroxy-4′methoxyflavanone-7-O-β-glucopyranosyl-(1→6)-β-rhamnoside.

6) In the introduction, the authors should include a figure presenting the structure of NEO

Answer

In the present version, structure of Neoponcirin is presented in the Introduction section.

7) In the introduction, it should be clearly stated if there were any previous articles describing the complexes of NEO/βCD, i.e. [13], and what is currently known about such complexes

Answer

Reference 13 is about flavonoid complex of rutin, to our best knowledge there is not previous reports of NEO Inclusion complexes.

8) Line 118, here, after such statement, it would be beneficial to include a reference to a review on NMR analysis of cyclodextrin complexes

Answer

We have included a reference:

https://www.researchgate.net/publication/221926769_Review_Cyclodextrin_Inclusion_Complexes_Probed_by_NMR_Techniques

9) Table 1, chemical shifts should be rounded at most to 3 decimal places, due to the accuracy of this method

Answer

This was fixed.

10) Figure 2A, spectrum of Beta-CD should be added to enable the proper comparison

Spectrum of bCD is the first spectrum in the figure.

Answer

This was modified.

11) Figure 3, the dissolution should be presented in mol/dm3 to show if really higher concentration of NEO can be achieved by the used of beta-CD.

Answer

This was corrected.

12) Line 469, it should be “NMR” not “NM”

Answer

This was fixed.

13) Lines 476-477, the details of FT-IR and UV-VIS analysis must be provided. I.e., the spectrometers used, resolution, mode (KBr? ATR?), similarly to how the

Answer

This was modified. Fourier transform infrared (FT-IR) spectra were recorded on a Thermo Scientific Nicolet 6700, in ATR mode and the scanning range was 450−4000 cm−1 with thirty-two scans. UV−Vis spectral analysis was performed using a VELAB-5100UV spectrophotometer with a quartz cell with a path length of 1 cm.

14) Lines 516-518, this sentence is incomplete

Answer

This was fixed.

15) Lines 656-661, this part should be removed

Answer

This was fixed.

16) Table 2, the values are not rounded properly, i.e. it should be 51.87 ± 4.90

Answer

This was fixed.

17) Lines 308-309, this is not entirely true, the complexes don’t possess better absorption as CD is not being absorbed but metabolized. The formation of complexes

Answer

This imprecision was corrected in the present version. We explain that βCD can facilitate the molecule to reach its pharmacological target in a biological system.

18) Lines 324-326, this statement about enantioselectivity is based on exactly which results?

Answer

In our experience with this molecule, we have observed that only the S epimer is encapsulated when we prepared the mixture of R and S (formed when pH is lightly acid or as an artifact during the isolated process), leaving a remnant of R free, which is not complexed.

However, you are right. In this study, we do not provide information to support that. We must address specific studies aimed at supporting this idea. We delete this assertion. Thank you very much for your observation.

Sincerely,

 Rosa Estrada-Reyes

Laboratorio de Fitofarmacología

Instituto Nacional de Psiquiatría Ramón de la Fuente Mucñiz

Ciudad de México, Mexico

Round 2

Reviewer 2 Report

Comments and Suggestions for Authors

While I appreciate that the Authors have answered my minor comments accordingly, two main aspects have not been addressed properly. Therefore, I will repeat them below:

1) I don’t understand why the authors haven’t used a positive control group in which the NEO would be administrated at the same dosage, but non complexed, without CD. It is reasonable to have a negative control group with CDs and the authors have done this. But what about the positive control? After all, the NEO should work as the CD doesn't have the desired biological activity. The Authors postulate, that the complexation with CD has the beneficial effect. Therefore, for comparision, the "pure" non-complexed NEO must be used.

2) The other aspect I’m very worried about is the dissolution analysis. The Authors have lyophilized the obtained complexes, which led to the amorphization. It is well known that the amorphous (lyophilized) APIs are characterized by more rapid dissolution. On the other hand, NEO used for comparison was not lyophilized.

Therefore, the Authors don’t know whether the increase in the dissolution rate was caused by the complexation or lyophilization or was the result of those both factors. To be honest, the Authors should perform two more dissolution studies: one with crystallized complexes (not subjected to lyophilization) and one with lyophilized NEO.

The Authors write, in their response to my previous comments, "The evidence obtained through NMR, UV-Vis, and FT-IR presented in this article supports the formation of a complex between neoponcirin and cyclodextrin, which positively impacts their solubility.". However, the complexation doesn't always increase the solubility so simply the fact that the complexes are formed doesn't indicate that the solubility has increased. Therefore, my previous comment is still valid.

If the Authors can't or don't want to address those comments, I'm afraid I'll recommend the rejection of this incomplete work.

Round 3

Reviewer 2 Report

Comments and Suggestions for Authors

I appreciate the efforts of Authors to answer my questions and correct their manuscript. Current version can be accepted for publication.